# Development of a Mouse Experimental System for the In Vivo Characterization of Bioengineered Adipose-Derived Stromal Cells

**DOI:** 10.3390/cells13070582

**Published:** 2024-03-27

**Authors:** Taeko Ichise, Hirotake Ichise, Yusuke Shimizu

**Affiliations:** 1Department of Plastic and Reconstructive Surgery, University of the Ryukyus Hospital, 207 Uehara, Nishihara 903-0215, Japan; h184093@med.u-ryukyu.ac.jp (T.I.); yyssprs@gmail.com (Y.S.); 2Institute for Animal Research, Faculty of Medicine, University of the Ryukyus, 207 Uehara, Nishihara 903-0215, Japan; 3Department of Plastic and Reconstructive Surgery, Graduate School of Medicine, University of the Ryukyus, 207 Uehara, Nishihara 903-0215, Japan

**Keywords:** adipose-derived stromal cells, mouse, immortalization, transplantation

## Abstract

Human adipose-derived stromal cells (ADSCs) are an important resource for cell-based therapies. However, the dynamics of ADSCs after transplantation and their mechanisms of action in recipients remain unclear. Herein, we generated genetically engineered mouse ADSCs to clarify their biodistribution and post-transplantation status and to analyze their role in recipient mesenchymal tissue modeling. Immortalized ADSCs (iADSCs) retained ADSC characteristics such as stromal marker gene expression and differentiation potential. iADSCs expressing a fluorescent reporter gene were seeded into biocompatible nonwoven fabric sheets and transplanted into the dorsal subcutaneous region of neonatal mice. Transplanted donor ADSCs were distributed as CD90-positive stromal cells on the sheets and survived 1 month after transplantation. Although accumulation of T lymphocytes or macrophages inside the sheet was not observed with or without donor cells, earlier migration and accumulation of recipient blood vascular endothelial cells (ECs) inside the sheet was observed in the presence of donor cells. Thus, our mouse model can help in studying the interplay between donor ADSCs and recipient cells over a 1-month period. This system may be of value for assessing and screening bioengineered ADSCs in vivo for optimal cell-based therapies.

## 1. Introduction

Mesenchymal stromal cells (MSCs) are self-renewable progenitor cells that can differentiate into mesenchymal lineage cells, such as adipocytes, osteocytes, and chondrocytes. MSCs’ secretion of growth factors, cytokines, extracellular matrices, and extracellular vesicles also supports and modulates the proliferation, motility, and function of neighboring cells in a paracrine manner. Extensive studies have shown that MSCs, MSC-conditioned media, and MSC-derived extracellular vesicles promote tissue regeneration and wound repair when transplanted or infused into animals and humans and that the MSC-derived secretome is deeply involved in homeostatic processes. The therapeutic efficacy of MSCs has been enthusiastically investigated in preclinical and clinical trials for treating diseases, as well as in vivo experiments using animal models [1].

However, MSC-based therapies in basic and preclinical research often experience problems and failures [1]. Large doses of MSCs are usually necessary for disease treatment because of their short residence time and/or suboptimal function following transplantation. The exposure of transplanted MSCs to environmental stresses in recipient tissues, particularly under pathological conditions, and their response to this reduce their viability and may alter their characteristics. Primary MSCs undergo cellular senescence during ex vivo culture for cell expansion. Aged MSCs may therefore exhibit a senescence-associated secretory phenotype and lose their original cellular characteristics and secretome, producing suboptimal and unwanted effects in recipients [2].

Moreover, transplanted MSCs are potential candidates for immune rejection. MSCs are widely believed to be immune-privileged. This concept is supported by results of low-level MHC expression by MSCs, immunomodulation of immune cells by MSCs in vitro, anti-inflammatory effects of transplanted MHCs into inflammatory tissues, and successful transplantation of allogeneic MSCs. However, subsequent immune responses can eliminate transplanted MSCs [3], and ex vivo-cultured, syngeneic MSCs were recently reported to induce a local innate immune response when transplanted into immunocompetent mice [4].

In addition to the paracrine effects of living MSCs, recent studies have found that the efferocytosis of intravenously administered MSCs, which involves the clearance of apoptotic MSCs by recipient phagocytes, positively regulates anti-inflammatory responses in animal models of disease [5,6]. Such efferocytosis-dependent therapeutic effects require dying MSCs rather than living MSCs. Nevertheless, many reports have shown that cell encapsulation and cell aggregation, both of which facilitate prolonged cell survival and retention, enhance the therapeutic effects of transplanted MSCs [1]. Moreover, prolonged survival of subcutaneously transplanted MSCs reduced skin fibrosis and inflammation in mice and rats [7,8].

Cell and genetic engineering technologies, such as three-dimensional cell cluster formation, ligand priming, and gene editing, have been recently used to “design” MSCs to improve the efficacy of MSC-based therapies [1]. To assess such designed MSCs, basic information about the transplanted MSCs in recipients is crucial. The biodistribution and residence time of transplanted MSCs have been studied using cell trackers, transgenic reporters, and in vivo macroimaging systems. However, the characteristics and roles of transplanted MSCs in situ remain poorly understood, and the precise mechanism of tissue regeneration after MSC transplantation is unclear.

Herein, we focused on adipose-derived stromal cells (ADSCs), which are MSCs found in adipose tissue. Human adipose tissue is routinely removed during surgical procedures and treated as medical waste. Human ADSCs have been isolated from such clinical materials, expanded, and stored as bioresources for research and development. To gain more insight into using ADSCs for this purpose, we sought to develop a simple and conventional method for tracking transplanted ADSCs in a mouse experimental system. We generated immortalized mouse ADSCs that expressed a fluorescent reporter gene. These cells were then transplanted via biocompatible nonwoven fabric sheets into neonatal mice that harbored a permissive environment for transplanted cell engraftment. Using this system, we observed blood vascular endothelial cell (EC) recruitment by transplanted ADSCs in an in vivo microenvironment.

## 2. Materials and Methods

### 2.1. Mice

All mice used in this study were derived from a closed colony of CGE transgenic mice [9] with a B6J;129S mixed background. CGE transgenic mice harboring a Cre driver allele express β-geo before Cre recombination and blasticidin-resistant enzymes and EGFP after Cre recombination. All mice used as ADSC donors and recipients had the transgene but no Cre driver allele. Transgenic reporters or drug-resistant enzymes expressed in CGE mice were not used in the study. Mice were housed under pathogen-free conditions. All animal experiments were approved by the University of the Ryukyus (approval nos. A2022057, A2023024) and were conducted in accordance with their guidelines.

### 2.2. Cell Culture

Primary ADSCs were isolated from 3- to 4-week-old female CGE transgenic mice. Mouse subcutaneous fat pads were dissected, washed with phosphate-buffered saline (PBS), diced, and dispersed in 2 mg/mL type I collagenase at 37 °C for 30 min. Dulbecco’s modified Eagle medium containing 10% fetal calf serum was then added to stop the enzyme reaction, and undispersed tissue was removed using a cell strainer. The cell suspension was centrifuged at 440× *g* for 5 min, and the supernatant was removed. Cells were washed with PBS, and supernatant removal via centrifugation, repeated twice. Finally, cell pellets were dispersed with a MesenCult Expansion Kit (Mouse) (STEMCELL Technologies, Vancouver, BC, Canada) supplemented with penicillin–streptomycin–amphotericin B suspension (100×) (FUJIFILM Wako Chemicals, Osaka, Japan), plated on cell culture dishes, and incubated at 37 °C in a humidified incubator with 5% CO_2_. Cell suspensions of approximately 1 g of fat pads from five female mice were plated on a 10 cm culture dish.

### 2.3. Plasmid Construction

mTert-pBabe-puro (Addgene plasmid #36413) (Addgene, Watertown, MA, USA), pMXs-gw (Addegene plasmid #18656), and pBRPy-nuclear mCherry-IRES-Puro (Addegene plasmid #52409) were gifts from Dr. Marta Alvarez and Dr. Joseph Bidwell, Dr. Shinya Yamanaka, and Dr. Jacob Hanna, respectively. The original mTert-pBabe-puro harbors cDNA identical to BC127068.1 (NM001362388), which encodes a putative N-terminal–truncated mTert protein [10]. We constructed a modified mTert-pBabe-puro (mTert^FL^-pBabe-puro) containing cDNA identical to NM009354. The resulting cDNA encodes a full-length mTert highly homologous to hTERT, which has been frequently used for cell immortalization. The cDNA fragment for replacement was PCR-amplified using E14.1 mouse embryonic stem cell cDNA and sequence-verified. The SV40tsA58 large T antigen (Ag) cDNA-harboring plasmid was previously described [11]. cDNA was subcloned into pMSCV-hygro (Clontech/TaKaRa Bio, Shiga, Japan). pMX-Bsd-mCherry-nls was constructed by removing an attR1-attR2 gateway cassette from pMXs-gw and inserting mCherry-nls cDNA, followed by the SV40-EM7-Bsd cassette (Invitrogen, Carlsbad, CA, USA). mCherry-nls cDNA was derived from pBRPy-nuclear mCherry-IRES-Puro.

### 2.4. Retroviral Transduction

Ecotropic retroviral vectors for the expression of mTert, tsA58T Ag, and mCherry-nls in ADSCs were produced by retroviral plasmid transfection into Plat-E packaging cells (a gift from Dr. Toshio Kitamura) [12] with Fugene HD Transfection Reagent (Promega, Madison, WI, USA). At 24 h after transfection, the culture medium was replaced with fresh medium, and the virus-containing supernatant was collected 16 h later for infection. The filtered supernatant was supplemented with 8 µg/mL polybrene (for ADSCs). For infection, the culture medium was replaced with virus-containing supernatants, and the culture plates were centrifuged (1 h, 1500 rpm). Plates were then infected for 8 h. At 48 h after infection, cells were selected for a week in 1.5 µg/mL puromycin (for mTert), 200 µg/mL hygromycin (for tsA58T Ag), and 5 µg/mL blasticidin (for mCherry-nls).

### 2.5. Immortalization of Primary ADSCs

Primary mouse ADSCs cultured on day 5 were infected with retroviruses harboring mTert^FL^-pBabe-puro to generate ADSCs-mTert. After puromycin selection and cell expansion, ADSCs-mTert cultured on day 27 were infected with retroviruses harboring pMSCV-hygro-tsA58T Ag to generate immortalized ADSCs (iADSCs). Hygromycin selection and culture at 33 °C (a permissive temperature for thermolabile tsA58T Ag protein) were initiated 2 days after retrovirus infection. A puromycin- and hygromycin-resistant pooled cell population that continued to proliferate represented iADSCs in this study.

### 2.6. WST-1 Proliferation Assay

Primary ADSCs on days 6 and 13 of culture and iADSCs on day 64 of culture were seeded at 3.0 × 10^3^ cells per well in a 96-well plate and cultured in ADSC-BulletKit (Lonza, Basel, Switzerland). Cell proliferation was analyzed using the Premix WST-1 Cell Proliferation Assay System (TaKaRa Bio, Shiga, Japan) for 4 days.

### 2.7. Differentiation Assays

ADSCs and iADSCs were seeded at 3.2 × 10^4^ cells per well in a Nunc cell-culture- treated 4-well dish (ThermoFisher Scientific, Waltham, MA, USA). After 2 days, the adipogenic differentiation was induced by incubation in a MesenCult Adipogenic Differentiation Kit (Mouse) (STEMCELL Technologies). After 4 and 7 days, cells were stained with Oil Red O (Sigma-Aldrich, St Louis, MO, USA) to determine the presence of intracellular lipid droplets, and the expression of differentiation markers after 7 days of differentiation was examined using real-time quantitative PCR (RT-qPCR). Osteogenic differentiation of ADSCs and iADSCs was induced by incubation in a MesenCult Osteogenic Stimulatory Kit (Mouse) (STEMCELL Technologies) for 11–21 days. Cells after 21 days of differentiation were then stained with Alizarin red (MUTO PURE CHEMICALS, Tokyo, Japan) to detect Ca^2+^ deposits. Cells after 11–21 days of differentiation were also examined for expression of differentiation markers via RT-qPCR. For Alizarin red staining, Nunc cell-culture-treated 4-well dishes (ThermoFisher Scientific) were coated with 0.01% type I collagen (C4243, Sigma-Aldrich).

### 2.8. RT-qPCR

Total RNA extracted from ADSCs and iADSCs with TRIzol reagent (Invitrogen) was treated with DNase I and purified. cDNA was prepared using SuperScript VILO Master Mix (Invitrogen). RT-qPCR was performed using the StepOne real-time PCR system with the PowerUp SYBR Master Mix (ThermoFisher Scientific) and specific primer sets. Actb was used as an endogenous control transcript for normalization and quantification of gene expression. Primer sequences are shown in Table 1. Samples were run at least in triplicate and analyzed using the comparative Ct method.

### 2.9. Preparation of 3D iADSC Sheets

Biocompatible nonwoven fabric sheets (ORTHOREBIRTH, Yokohama, Japan), which are made of fibers comprising biodegradable poly(lactic-co-glycolic acid) and hydroxyapatite [13], were cut into 3 mm squares and placed one per well in a Nunclon Sphera 96F Bottom Plate. After each well was filled with ADSC-BulletKit (Lonza), plates were degassed in a vacuum desiccator to remove air bubbles from the sheets and then preincubated for 1 h in a humidified incubator with 5% CO_2_. iADSCs-mCherry-nls (4 × 10^4^) cells were added per well and cultured at 33 °C for 7–14 days. Sheets harboring the iADSCs were then removed and dipped into Matrigel Matrix High Concentration (Corning, Corning, NY, USA) before transplantation. Resulting sheets are referred to as 3D iADSC sheets. For the negative control, fabric sheets were degassed, precultured, and dipped into Matrigel without cell culture.

### 2.10. Scaffold-Assisted Transplantation of iADSCs into Neonatal Mice

CGE transgenic neonatal mice (both sexes) on postnatal days 1–2 were used as recipients. Neonatal mice were placed on a nitrile glove in an ice bath to induce hypothermia anesthesia, which is appropriate for short, minor surgical procedures [14]. All procedures were performed in a laminar flow cabinet. A 5–10 mm transverse incision was made in the skin on the mouse back, and the subcutaneous tissue was bluntly dissected for subcutaneous transplantation of 3D iADSC sheets. One 3D iADSC sheet was introduced into the subcutaneous space of a neonate. The incised skin was sutured, and the incision dressed with adhesive plasters to prevent the mother mice from removing sutures while nursing for several days. The entire operation for one neonate, from anesthesia to dressing, was performed within 10 min. Postoperated mice were kept warm until waking and then returned to their mother mice.

### 2.11. Fluorescence Live-Cell Imaging and Immunohistochemistry

Sheets and surrounding tissues were removed at 4, 8, 16 days and 1 month after cell transplantation. Flat-mount fluorescent images of the sheets were acquired with a BioRevo 9000 microscope (Keyence, Osaka, Japan), and the sheets were then fixed in 4% paraformaldehyde in PBS at 4 °C overnight. Fixed specimens were placed in acetone at −20 °C overnight to dissolve solid biodegradable fibers, rinsed in PBS, dehydrated in PBS containing 20% sucrose for 4 h, and then processed for cryosectioning. Frozen sections (5 μm) were washed in PBS, and antigen retrieval was accomplished by incubation in PBS containing 0.125% trypsin and 0.5 mM EDTA at room temperature for 20 min. All sections were incubated in methanol containing 3% H_2_O_2_ at room temperature for 20 min before incubation with primary antibodies. Sections were incubated with a primary antibody, followed by Histofine reagent (Nichirei Biosciences, Tokyo, Japan), and immunostaining visualized with the TSA PLUS HRP Detection System (NEN/ Perkin-Elmer, Waltham, MA, USA) and DAPI (Molecular Probes/Invitrogen, Carlsbad, CA, USA). For immunostaining for F4/80, sections were incubated with primary antibody (rat antimouse F4/80) and secondary antibodies labeled with AlexaFluor 488 (Molecular Probes/Invitrogen) for visualization. For double immunostaining, after immunostaining with the first antibody and visualization, sections were incubated with 3% H_2_O_2_ to quench the peroxidase activity of the Histofine reagent. Sections were then incubated with secondary primary and secondary antibodies. The following primary antibodies were used: rabbit anti-mCherry (Proteintech, Rosemont IL, USA); rat antimouse CD90.2 (Biolegend, San Diego, CA, USA); rat antimouse VEGFR-2 (eBioscience, San Diego, CA, USA); rat antimouse F4/80 (eBioscience); rat antimouse CD3 (Proteintech); and rabbit anti-Perilipin 1 (Proteintech). For some panels, frozen sections were stained with hematoxylin and eosin (H&E) stains. Fluorescence micrographs were acquired using a BioRevo 9000 microscope (Keyence). Measurement of immunostained and DAPI-stained areas was performed using the measurement module of the microscope software for BZ-9000 (Keyence). Micrographs in the figures are representative of two independently stained specimens from three or more mice.

## 3. Results

### 3.1. Establishment and Characterization of Immortalized Mouse ADSCs

We first sought to use primary mouse ADSCs transduced with retroviral vectors for cell transplantation and tracking experiments; however, these cells were difficult to handle in subsequent experiments because they stopped proliferating as the culture time and passage number increased with antibiotic selection of proviruses and cell expansion. To address this problem, we used a cell immortalization technique based on a previous study [15], which showed that the ectopic expression of both the catalytic subunit of human telomerase (hTERT) and the mutated large T Ag of a temperature-sensitive mutant of SV40 (tsA58) immortalize primary cells without changing cellular characteristics. The proliferation of nontransduced primary ADSCs slowed after 14 days of culture, and the cells died after 21–25 days of culture (7 to 8 passages). However, drug-resistant bulk ADSCs transduced with retroviral vectors harboring a mouse ortholog of hTERT, mTert, and tsA58 T Ag (hereafter, iADSCs) sustained proliferation for over 65 days of culture (Figure 1A). The expression of proliferation marker genes Mki67 (marker of proliferation Ki-67) and Cdk4 was maintained or slightly increased in iADSCs but decreased in primary ADSCs with increasing culture time (Figure 1B). On the other hand, the expression of senescence-associated secretory phenotype (SASP) marker genes was low in iADSCs (Figure 1C). The expression of undifferentiated MSC marker genes was unaffected in iADSCs (Figure 1D). These results indicated the successful immortalization of mouse primary ADSCs.

We next examined whether iADSCs could still differentiate into adipocytes and osteoblasts. As demonstrated by mRNA expression levels and cell type-specific staining (Figure 2), iADSCs differentiated into adipocytes and osteoblasts following the appropriate differentiation stimuli. In contrast to a previous report showing that the wild-type SV40 T Ag compromised the adipogenic and osteogenic differentiation of hTERT-overexpressing human ADSCs [16], the mutated type of SV40 T Ag did not affect mTert-overexpressing mouse ADSCs. These results show that iADSCs acquired prolonged cell survival in vitro without losing their self-renewal and differentiation capacity.

### 3.2. Transplantation of iADSCs into Neonatal Mice and Cell Tracking

iADSCs were transduced with an mCherry-expressing retroviral vector to enable tracking. In our preliminary study, ADSCs expressing cytosolic or membrane-bound fluorescent reporter proteins produced fluorescent reporter protein-positive extracellular vesicles in vitro (data not shown). To distinguish donor ADSCs from recipient cells that received or engulfed donor ADSC-derived extracellular vesicles in transplantation experiments, mCherry tagged with nuclear localization signal sequence peptides was used for ADSC labeling.

For the transplantation and engraftment of ADSCs and other MSCs, subcutaneously injection of ADSC suspensions have been shown to have poor survival time and rapidly decreased in number, regardless of allogeneic or syngeneic transplantation [1,4]. To improve cell viability and facilitate the retention of transplanted ADSCs, we chose commercially available, biodegradable nonwoven fabric sheets, which are reported to be suitable for maintaining ADSCs in vitro [13]. iADSCs-mCherry-nls were attached to the sheets by culturing cells with sheets in low-attachment cell culture plates. iADSC-containing sheets were coated with Matrigel to support cells in the absence of culture media during transplantation. We selected neonatal mice as recipients of iADSCs as these mice are considered ideal recipients for cell transplantation because the fast-growing and developing period of postnatal life may provide a developing niche and permissive environment for engrafting transplanted cells. Although the genetic backgrounds of the recipient mice and iADSCs were similar because they were both derived from the same closed colony of B6J;129S hybrid mice (H2-k^b^), mCherry-nls-expressing iADSCs expressed potential immunogens such as SV40 T Ag, antibiotic-resistant enzymes, and mCherry. Subsequently, we subcutaneously transplanted iADSC-containing fabric sheets (hereafter, iADSC 3D sheets) or cell-free fabric sheets into the back skin of newborn mice.

To examine the survival and duration of transplanted cells, 3D iADSC sheets were removed and observed for fluorescence 4, 8, and 16 days after transplantation. mCherry-nls-expressing ADSCs in the fabric sheets were present in seven of seven, six of six, and five of six mice examined at 4, 8, and 16 days after transplantation, respectively (Figure 3A). This result contrasts with the rapid clearance of the injected cell suspension within the first week in previous reports. iADSCs were shown to be resident as CD90-positive ADSCs in the fabric sheets (Figure 3B).

Immunostaining for mCherry showed that iADSCs remained viable 1 month after transplantation (Figure 4A). In the fabric sheets, iADSCs dispersed and did not form cellular nodules suggestive of hyperproliferation (Figure 4A). However, mCherry-nls-positive cells in the sheet were not found in two of the eight mice examined. We compared the iADSC-containing sheets at two time points, 4 days and 1 month after transplantation. The relative number of mCherry-nls-positive cells among cells in the 3D sheet 1 month after transplantation was lower than that 4 days after transplantation (Figure 4B). Although one reason for this is that recipient cells were migrating into the sheet, these results suggest that transplanted iADSCs do not proliferate rapidly in vivo and may be gradually lost during the weaning and post-weaning periods. Immunostaining of serial sections for mCherry, CD90, and Perilipin 1 as markers for transplanted iADSCs, stromal cells, and adipocytes, respectively, suggested that most iADSCs remained stromal cells but did not differentiate into adipocytes (Figure 4C).

Transplantation of syngeneic ADSCs has been reported to induce recipient immune responses and macrophage accumulation [4]. Consequently, we examined H&E-stained 3D sheet sections and found blood vessel formation but not leukocyte infiltration in the 3D iADSCs (Figure 5A). We also examined the distribution of macrophages and T lymphocytes in and around the fabric sheets. Although the adipose tissue adjacent to the fabric sheets 4 days after transplantation showed macrophage accumulation, the fabric sheets contained a fewer dispersed macrophages, regardless of the presence or absence of iADSCs (Figure 5B). T lymphocytes were rarely found in the fabric sheets and did not gather in or around the sheets (Figure 5C). These results suggest that iADSCs did not induce macrophage or T lymphocyte activation. Conversely, we found that VEGFR2-positive ECs were present in the iADSC-containing sheets 4 days after transplantation but not in the cell-free sheets (Figure 5D). Although ADSCs and MSCs can differentiate into ECs, double immunostaining showed that mCherry-nls-positive cells were negative for VEGFR2 (Figure 5D). This finding suggests that iADSCs recruit recipient ECs into fabric sheets.

Collectively, our results demonstrated that immortalized, trackable mouse ADSCs could be transplanted into neonates of immunocompetent mice and were resident for a month in biocompatible nonwoven fabric sheets in vivo (Figure 6). This mouse experimental system enables us to study the interplay between recipient cells and bioengineered donor ADSCs in a microenvironment in vivo.

## 4. Discussion

ADSCs and other tissue-derived MSCs are believed to exert therapeutic effects through a “hit-and-run” mechanism [1]; however, transplanted ADSCs have been repeatedly found to rapidly decrease in recipients, suggesting that their effects are brief and limited. Effective therapies require the transplantation/infusion of large numbers of ADSCs to compensate for this short life span and duration. To improve the weaknesses of ADSCs, studies have proposed cell and genetic engineering techniques, such as cell encapsulation, cell aggregate formation, genetic modification, and metabolic reprograming to ADSCs [1]. However, the effectiveness of such techniques in vivo has not been fully understood because of the lack of reliable in vivo experimental systems for studying ADSCs. The efferocytosis of intravenously administered MSCs contributes to the anti-inflammatory responses of recipient cells in the spleen and lungs [5,6]. In contrast, prolonged survival and retention of subcutaneously transplanted MSCs reduce skin fibrosis and inflammation [7,8]. In vivo experimental systems are needed to dissect the environment-dependent roles and effects of ADSCs.

To study the effect of human ADSCs in vivo, immunodeficient mice have been widely used as xenograft recipients. Severely immunodeficient animals allow the long-term survival of human ADSCs and can help in the examination of the impact of human ADSCs on recipient tissues, which is valuable in translational medicine. However, in most cases, ADSC-based therapeutics are used for allogeneic transplantation in immunocompetent patients. Some effects of human ADSCs can be studied by xenograft modeling using immunocompetent mice [17], but attention must be paid to the interplay between donor ADSCs and recipient immune cells, which may have a crucial role in the fate and function of ADSCs and their anti-inflammatory effects in an allogeneic environment [18]. Transplanted ADSCs in clinical settings are exposed to severe environments such as damaged, swollen, avascular, and/or inflammatory tissues, and such cellular stress responses do not always activate cell survival pathways but instead initiate cell death programs. Cell dissociation before transplantation may evoke cellular stress responses in anchorage-dependent ADSCs. Transplanted ADSC suspensions have been reported to rapidly decrease in number even when transplanted into syngeneic mice [4]. In addition, MSCs undergo cellular senescence during cell culture in vitro and systematic aging of whole bodies in vivo, causing the loss of original MSC characteristics and subsequent cell death [2]. These findings prompted us to improve the microenvironment around ADSCs and the intrinsic activities of ADSCs against cellular senescence.

Therefore, we applied three techniques, immortalization, cell tracking, and scaffold-assisted cell engraftment, to efficiently handle and study ADSCs. We generated immortalized mouse ADSCs for transplantation into immunocompetent mice. A thermolabile mutant of the SV40 large T Ag and the TERT gene product are widely used for cell immortalization or prolonged cell lifespan without causing cell transformation or losing the original cellular characteristics [15]. Tumorigenesis is a major concern when immortalized ADSCs are used in vivo. In previous studies, immortalized MSCs, which express hTERT along with SV40 T Ag, could not form tumors when transplanted into immunodeficient mice, although immortalized cells overexpressing an additional cancer driver gene formed sarcomas [19]. However, the addition of spontaneous driver mutations or epigenetic alteration in endogenous genes during long-term experiments may cause unwanted phenotypes of iADSCs. The iADSCs in the current study were not tumorigenic in our experimental setting, and we therefore recommend this system for assessing designed ADSCs in experiments conducted a few months after transplantation. To refine our experimental system, controllable methods of genetic and cell manipulation, such as tunable gene expression systems of endogenous genes involved in cellular senescence and immortalization [20], a well-refined HSV-TK suicide gene system [21], and a transgene-free safety switch by metabolic engineering [22], would help circumvent the potential problem of hyperproliferation and tumorigenesis. The conventional retroviral transduction used in this study is time- and cost-effective in performing experiments and more suitable for high-throughput screening. This could be replaced with a more refined technique, such as CRISPR-mediated gene editing, in the next phase of studies that focus on translational medicine [1,17].

One of the most convenient methods for transplantation or infusion of ADSCs is injection of cell suspension. However, the viability of injected MSCs in suspension is less than that of cells transplanted as cell sheets, spheroids, and scaffolds [1]. Cell suspension injection is adequate for the systemic administration of cells via blood circulation but not for the local and sustained duration of injected cells. Herein, we showed that using biocompatible nonwoven fabric sheets as scaffolds supported the viability of donor ADSCs for a month. We propose that the sheets are suitable as scaffolds for ADSC transplantation into subcutaneous areas.

In our experimental system, transplanted ADSCs remained resident in scaffolds for a month, allowing us to study the interplay between donor ADSCs and recipient cells in vivo, in contrast to a previous report where syngeneic MSCs had a short lifespan (7 days) in mice [4]. Infiltration and accumulation of T lymphocytes and macrophages into scaffolds with or without ADSCs were not observed for several weeks after transplantation. Therefore, the designed ADSCs and the biodegradable sheet remained nonimmunogenic for cellular immune responses during neonatal development. The neonatal and postnatal immune systems are unique and different from the adult immune system. These systems recognize and respond to antigen exposure after birth. However, the systems establish tolerance to self-, microbiota-, diet-, and environment-derived antigens from birth to post-weaning while acquiring immune homeostasis and allowing microbial colonization after birth [23]. Thus, neonatal mice may be prone to tolerating iADSCs that express exogenous proteins.

Conversely, we found that ADSCs recruited ECs into the scaffolds during the first week after transplantation. ADSCs and other MSCs are believed to induce angiogenesis because they induce cellular proliferation and cord-like structure formation of ECs in vitro, and transplanted MSCs reduce ischemic diseases in vivo [24]. However, whether MSCs participate in the reduction of ischemic diseases through the direct induction of angiogenesis remains unknown. Our findings suggest that ADSCs resident in the avascular area induce angiogenesis by recruiting ECs, but not macrophages, which also stimulate angiogenesis. Previous studies reported that MSCs can transdifferentiate into multilineage cells, including ECs, in vitro [25,26]; however, such a cellular change was not observed in our in vivo study. These findings demonstrate that our experimental system is useful for understanding the interplay between donor ADSCs and recipient cells. Thus, our system will be useful for in vivo proof of concept studies of ADSCs designed for optimal cell-based therapeutics.

ADSC design for transplantation into disease mouse models is important to support progress in basic research for translational medicine. In most cases, transplanted ADSCs are expected to stimulate tissue regeneration and repair under pathological conditions rather than normal tissue modeling under physiological conditions. As differences are present in tissue development, homeostasis, and immune responses between neonates and adults, transplantation of the iADSC 3D sheets into adult mice and disease mouse models is necessary for improved understanding of ADSCs in cell-based therapies.

## Figures and Tables

**Figure 1 cells-13-00582-f001:**
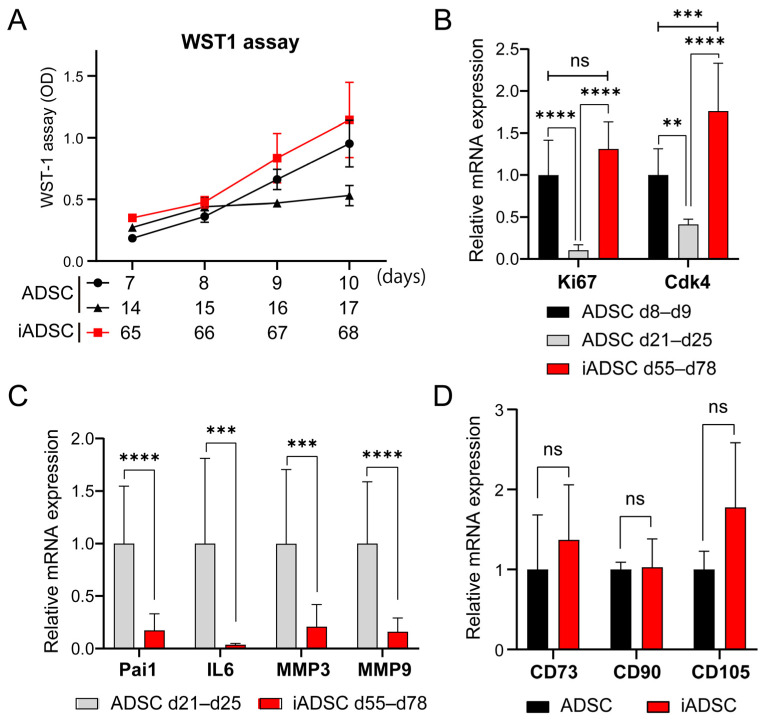
Phenotypic characterization of iADSCs. (**A**) WST-1 assay. Growth of primary cultured ADSCs after 2 and 3 weeks of culture and iADSCs after 2 months of culture were examined using the WST-1 assay. Error bars, s.d., *n* = 10. (**B**) Expression levels of proliferation markers in primary cultured ADSCs (cells on day 8–9 and 21–25 of culture) and iADSCs (cells on day 55–78 of culture). Error bars, s.d., *n* = 9. Statistical analysis, two-tailed mixed ANOVA and Tukey’s test. ** *p* < 0.01; *** *p* < 0.005; **** *p* < 0.001. (**C**) mRNA expression levels of senescence-associated secretory phenotype (SASP) marker genes in primary cultured ADSCs (cells on day 21–25 of culture) and iADSCs (cells on day 55–78 of culture). Error bars, s.d., *n* = 9. Statistical analysis, two-tailed Student’s *t*-test. *** *p* < 0.005; **** *p* < 0.001. (**D**) mRNA expression levels of mesenchymal markers in primary cultured ADSCs (cells on day 9 of culture) and iADSCs (cells on day 52–55 of culture). Error bars, s.d., *n* = 6. Statistical analysis, two-tailed Student’s *t*-test. ns, not significant.

**Figure 2 cells-13-00582-f002:**
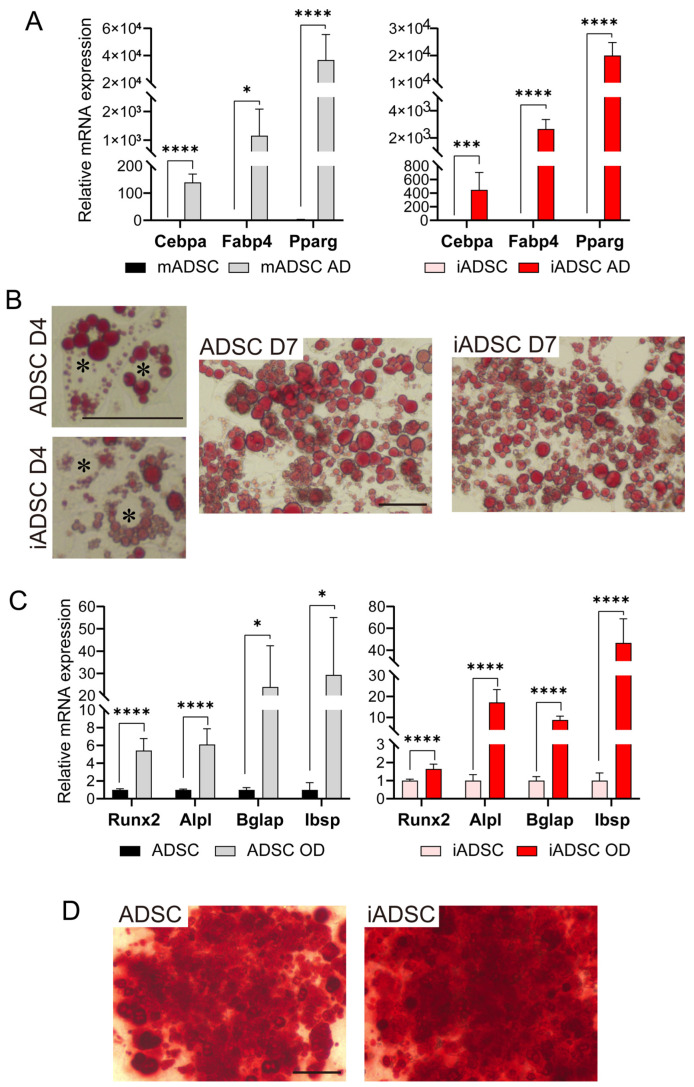
Investigation of the differentiation capacity of iADSCs into adipocytes and osteoblasts. (**A**) RT-qPCR assay for the expression of marker genes in ADSCs (**left**) and iADSCs (**right**), before and after adipogenic differentiation. Error bars, s.d., *n* = 6. Statistical analysis, two-tailed Student’s *t*-test. * *p* < 0.05; *** *p* < 0.005; **** *p* < 0.001. (**B**) Oil red O staining of ADSCs and iADSCs at 4 days and 7 days of adipogenic differentiation. Scale bar, 100 μm. Asterisks indicate the position of cell nuclei. (**C**) RT-qPCR assay for marker genes in ADSCs (**left**) and iADSCs (**right**), before and after osteogenic differentiation. Error bars, s.d., *n* = 6. Statistical analysis, two-tailed Student’s *t*-test. * *p* < 0.05; **** *p* < 0.001. (**D**) Alizarin red staining of ADSCs and iADSCs after induction of osteogenic differentiation for 21 days. Scale bar, 100 μm.

**Figure 3 cells-13-00582-f003:**
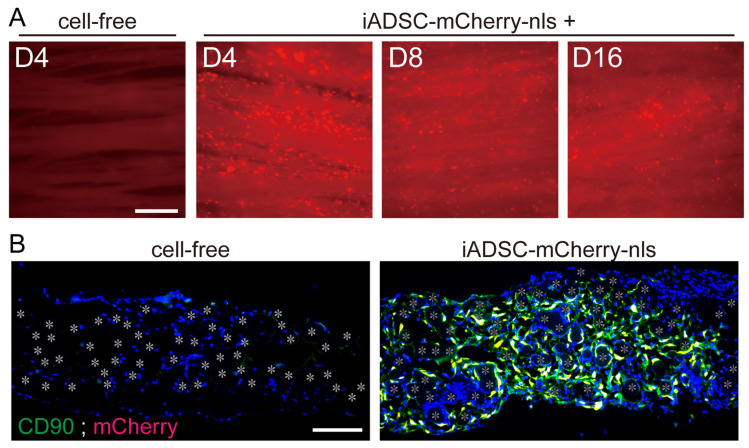
Survival of transplanted iADSCs in biodegradable fabric sheets during the postnatal period. (**A**) Live imaging of 3D iADSC sheets after transplantation. At 4, 8, and 16 days after transplantation, 3D iADSC sheets (**right**) were removed from the recipients and observed for fluorescence. As a negative control, a cell-free sheet is shown 4 days after transplantation (**left**). Scale bar, 200 μm. (**B**) Double immunostaining of cell-free (**left**) and 3D iADSC (**right**) sheets for mCherry and CD90 4 days after transplantation. Red: mCherry; green: CD90; blue: DAPI. Scale bar, 100 μm. The nuclei of double-positive cells are presented in yellow. Asterisks indicate the position of the fibers in the biodegradable sheet.

**Figure 4 cells-13-00582-f004:**
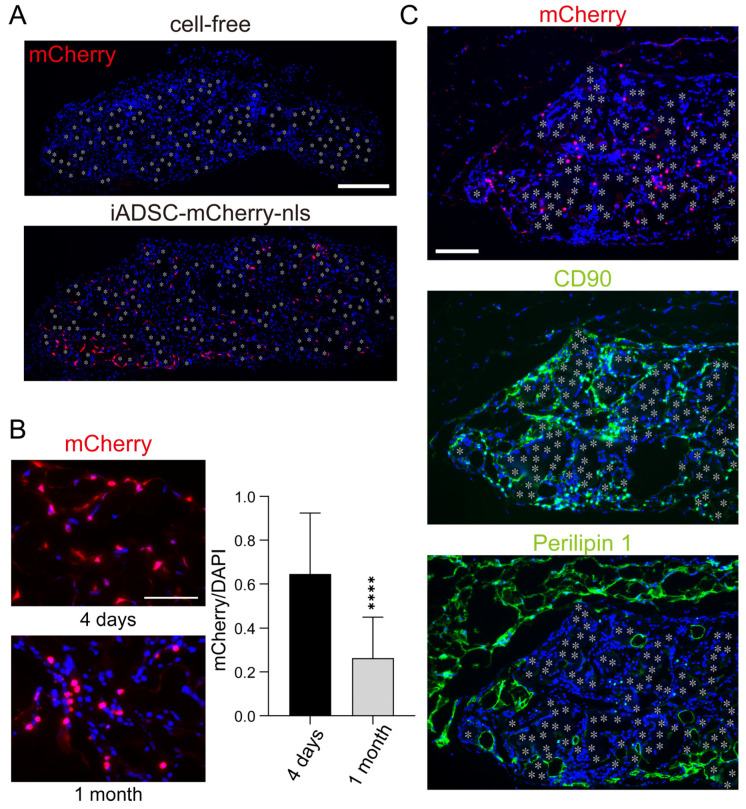
Detection and characterization of iADSCs in biocompatible fabric sheets 1 month after transplantation. (**A**) Immunostaining for mCherry in cell-free and 3D iADSC sheets 1 month after transplantation. Red: mCherry; blue: DAPI. Scale bar, 200 μm. (**B**) Immunostaining for mCherry in 3D iADSC sheets 4 days and 1 month after transplantation. Scale bar, 50 μm. Quantification of mCherry-immunostained areas normalized to DAPI-stained areas in 3D iADSC sheets is shown in the right-hand graphs. Error bars, s.d., *n* = 16 (four independent areas from four sheets harboring mCherry-positive cells). Statistical analysis, two-tailed Student’s *t*-test. **** *p* < 0.001. (**C**) Immunostaining of near-serial sections of 3D iADSC sheets for mCherry (**left**), CD90 (**middle**), and Perilipin1 (**right**) at 1 month after transplantation. Red: mCherry; green: CD90 (**middle**) or Perilipin1 (**right**); blue: DAPI. Scale bar, 100 μm. Asterisks indicate the position of the fibers in the biodegradable sheet.

**Figure 5 cells-13-00582-f005:**
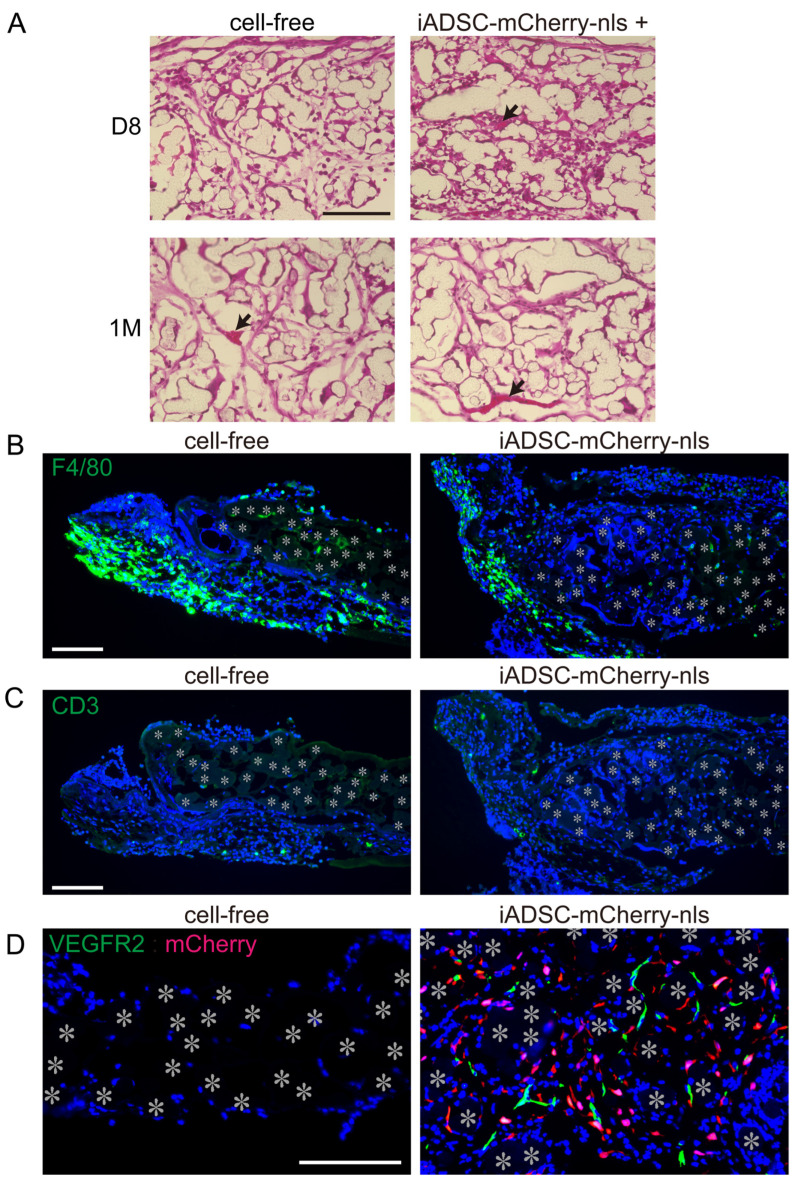
Effects of transplanted 3D iADSC sheets on recipient cells. (**A**) H&E staining of cryosections of cell-free and 3D iADSC sheets 8 days and 1 month after transplantation. Scale bar, 100 μm. Non-stained or weakly stained areas are traces of fibers in the biodegradable sheet dissolved in organic solvent. Arrows indicate blood vessels. (**B**) Immunostaining of cell-free (**left**) and 3D iADSC (**right**) sheets for F4/80 4 days after transplantation. Green: F4/80; blue; DAPI. Scale bar, 100 μm. (**C**) Immunostaining of cell-free (**left**) and 3D iADSC (**right**) sheets for CD3 4 days after transplantation. Green: CD3; blue: DAPI. Scale bar, 100 μm. (**D**) Double immunostaining of cell-free (**left**) and 3D iADSC (**right**) sheets for mCherry and VEGFR2 4 days after transplantation. Red: mCherry; green: VEGFR2; blue: DAPI. Scale bar, 100 μm. Asterisks indicate the position of the fibers in the biodegradable sheet.

**Figure 6 cells-13-00582-f006:**
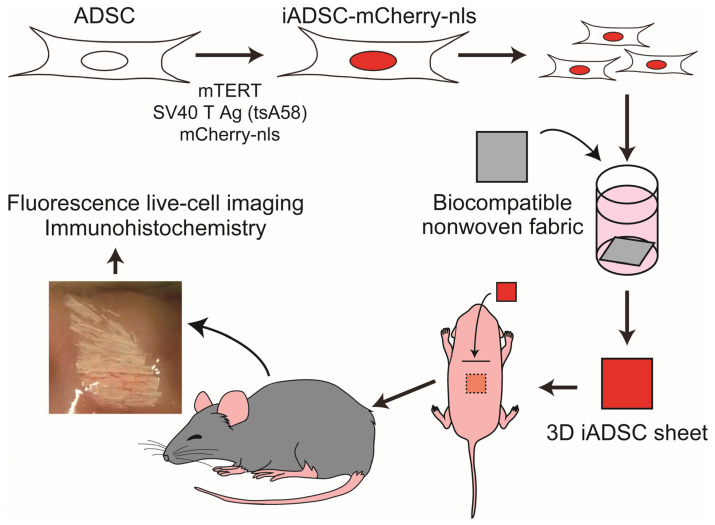
Schematic of the mouse experimental system for assaying ADSCs.

**Table 1 cells-13-00582-t001:** RT-qPCR Primers.

Name	Forward Primer	Reverse Primer
Ki67	AGGGTAACTCGTGGAACCAA	TTAACTTCTTGGTGCATACAATGTC
Cdk4	CCTCACGCCTGTGGTGGTTA	CCCAACTGGTCGGCTTCAGA
CD73	GCTTCAGGGAATGCAACATG	TGCCACCTCCGTTTACAATG
CD90	GAGTCCAGAATCCAAGTCGG	AGTCCAGGCGAAGGTTTTG
CD105	GTGTTCCTGGTCCTCGTTTC	TGTGATGTTGACTCTTGGCTG
Cebpa	AGAGCCGAGATAAAGCCAAAC	TCATTGTCACTGGTCAACTCC
Fabp4	GACAGGAAGGTGAAGAGCATC	GTCACGCCTTTCATAACACATTC
Pparg	TGTTATGGGTGAAACTCTGGG	AGAGCTGATTCCGAAGTTGG
Runx2	GCTATTAAAGTGACAGTGGACGG	GGCGATCAGAGAACAAACTAGG
Alpi	CTCCAAAAGCTCAACACCAATG	ATTTGTCCATCTCCAGCCG
Bglap	ACCATCTTTCTGCTCACTCTG	GTTCACTACCTTATTGCCCTCC
Ibsp	CCACACTTTCCACACTCTCG	CGTCGCTTTCCTTCACTTTTG
Pai1	TTGTCCAGCGGGACCTAGAG	AAGTCCACCTGTTTCACCATAGTCT
Il6	TCTCTGGGAAATCGTGGAAA	TCTGCAAGTGCATCATCGTT
Mmp3	GCTCATGCCTATGCACCTG	CATGAGCAGCAACCAGGAAT
Mmp9	TAGCTACCTCGAGGGCTTCC	GCTGTGGTTCAGTTGTGGTG
Actb	CGTGAAAAGATGACCCAGATCA	TGGTACGACCAGAGGCATACAG

## Data Availability

The datasets used and/or analyzed during the current study are available from the corresponding author upon reasonable request.

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
