# Peer review of "Development of a Mouse Experimental System for the In Vivo Characterization of Bioengineered Adipose-Derived Stromal Cells"

_cells, 2024, doi:10.3390/cells13070582_

Round 1

Reviewer 1 Report

Comments and Suggestions for Authors

The manuscript written by Ichise and colleagues, entitled "Development of a mouse experimental system for the in vivo characterization of bioengineered adipose-derived stromal cells," describes a murine model to study the distribution and interaction in immunocompetent models with the surrounding microenvironment. The paper may be of interest to the journal Cells; however, there are gaps that need to be addressed before publication. Therefore, I suggest a major revision.

MSCs are known for their immunomodulatory properties, although over the long term, they are reported to exert their action more through their paracrine action than through differentiation. That being said, as reported by the authors themselves, several articles mention that MSCs are killed or undergo apoptosis, and a few years ago, in a Science Translational Medicine article, Dazzi reported that MSCs can act through efferocytosis. These points should be mentioned in the introduction and discussed in the discussion section, especially why they should remain viable in their model. Furthermore, by using a neonatal system, I would also encourage the authors to discuss the aspect of immunocompetence from a young immune system to a mature one.

Regarding the experimental part, the authors refer to an article regarding the immortalization procedure and protocol, but they do not show results confirming the immortalization in their in vitro model. Referencing a study does not automatically mean that the obtained cells exhibit immortalization. Especially considering that the authors cite a study not from their group. In this regard, the authors should report the telomeric repeat amplification protocol (TRAP) assay to demonstrate the maintenance of telomere length. Up to which passage can the authors expand the primary cells? Have they performed senescence assays? It would be helpful to include bright-field morphological images as well as a beta-galactosidase assay.

Which strain of mice is used in the experimental protocol? Please specify. Onhow many mice the experiments was performed.

The authors should also perform immunofluorescence after one month post-transplantation and quantification, as well as on day 4 post-transplantation to evaluate viability variations. Additionally, it would be appropriate to include hematoxylin and eosin staining to visualize any necrotic or granulation tissue.

Comments on the Quality of English Language

The English is fine, only minor editing is required

Reviewer 2 Report

Comments and Suggestions for Authors

Dear authors,

I have several comments on your manuscript.

1. In Abstract and through the text your using combination 'blood vascular endothelial cells (BECs)' . It seems to be redundant? I think? 'endothelial cells (ECs)' would be enough. Moreover, this acronym is much more common than BECs.

2. In Materials and Methods (line 90) - what was the weight or the size of fat pads? How many cells were isolated from them and plated onto the dishes (line 98)? How many mice did you use do obtain ADSCs? How many independent cultures did you get?

How many cells were plated for the differentiation assay (line 145)? What type of culture dishes did you use?

3. Results. Figure 1 captions: why do you compare different culture types on different days of cultivation? Do you think it's adequate to compare, for example, the expression level of proliferation marker on day 8/21 in ADSCs, on day 23 in ADSCs-mTert and on day 78 in iADSCs? It seems to be much more representative, if you'd compare these parameters in all cultures on similar day, say day 21-23. If you aim to prove that there is no decrease in the proliferation with time in iADSCs, I would suggest to add a figure with the data on gene expression through the whole time of observation.

Next, the sample size of 3 is too small to establish any statistically significant effect. 

Figure 2B - Please, add the photos made with phase contrast to the photos of adipogenic differentiation. Otherwise there are just red spheres on the photo, without any proof that they are lipid drops inside adipocytes.

I think, in all experiments with transplantation the additional control of fabric sheets with non-transfected ADSCs would be useful. This way you would be able clearly demonstrate, that there is benefit in the survival and induction of vasculogenesis that is associated with genes' transduction and not with cells themselves. 

On line 345 you're writing that you've used immunocompetent mice, which is not entirely correct.

Round 2

Reviewer 1 Report

Comments and Suggestions for Authors

The authors ansered to all my concerns. The manuscript is acceptable for publication